

# AMMI and GGE biplot analysis for genotype × environment interactions affecting the yield and quality characteristics of sugar beet

Xinwang Dang[1], Xiaohang Hu[1,2], Yahuai Ma[1,2], Yanli Li[1,2], Wenliang Kan[3] and Xinjiu Dong[4]

[1] Academy of Modern Agriculture and Ecology Environment, Heilongjiang University, Harbin, Heilongjiang, China
[2] National Sugar Improvement Center, Harbin, Heilongjiang, China
[3] Heilongjiang Academy of Agricultural Reclamation Sciences, Qiqihar, Heilongjiang, China
[4] Cash Crop Research Institute of Xinjiang Academy of Agricultural Science, Urumqi, Xinjiang, China

Corresponding authors
Xiaohang Hu, 2005142@hlju.edu.cn
Wenliang Kan, kanwenliang@163.com

## ABSTRACT

Sugar beet, an important sugar crop, contributes significantly to the world's sugar production. However, genotype–environment interactions (GEI) often affect the quality characteristics of sugar beet. Hence, understanding the effects of GEI on sugar beet quality can aid in identifying high-quality genotypes that can adapt to different environments. Traditional variance analysis can only be used to examine the yield of a variety and not its specific adaptability to specific conditions. Therefore, more comprehensive analytical methods are required to evaluate the characteristics of the variety under specific environments. Additive main effects and multiplicative interaction (AMMI) and genotype main effect and genotype × environment interaction (GGE) biplot models can be employed to comprehensively evaluate different varieties and address the drawbacks associated with a single evaluation method. Moreover, these models also allow us to explore new varieties more objectively and comprehensively. In this study, the adaptability and stability of 16 sugar beet varieties, in terms of yield and sugar content, were evaluated using AMMI and GGE biplot analysis in seven pilot projects undertaken in 2022. In the assessment of a small but significant proportion of the total GEI variance for the two qualitative traits (yield and sugar content), 80.58% of the variance was explained by the cumulative contribution of IPC1, IPC2, and IPC3. AMMI and GGE biplots clearly highlighted that KWS4207 (G3) exhibited high and stable quality. They also demonstrated that the experiments in Jalaid Banner (Inner Mongolia) (E7) were the most representative. Together, the results suggested that the comprehensive application of AMMI and GGE biplot analysis allowed for a more comprehensive, scientific, and effective evaluation of sugar beet varieties across different regions. The findings offer a theoretical basis for sugar beet breeding and could guide the rational design of experiments for testing new varieties of sugar beet.

## INTRODUCTION

Sugar beet (*Beta vulgaris* L. biennial or the perennial herb of the genus *Chenopodium*) is an important cash crop and is used for industrial sugar production. Thus, it is of great significance from an economic development standpoint. In recent years, with China's economic growth, large-scale sugar beet cultivation has been undertaken in the northeast, north, and northwest of the country. Thus, sugar beet production has now become a key pillar industry (*Lu et al., 2023*), and increasing research on the quality of sugar beet varieties has improved the diversity of sugar beet breeding. In China, sugar beet development started only a few decades ago (*Ding et al., 2018*). However, promising breakthroughs have been made in several areas, including the breeding of disease-resistant varieties with immunity against common crop diseases, such as yellowing disease, root-rot disease, and root-cluster disease. This has significantly reduced the damage caused to sugar beet plants (*Ni et al., 2020*). Owing to continuously expanding production and the increasing demand for sugar beet, improving crop yield and quality has become a key challenge. Currently, regional variety tests are the most effective method for identifying and developing high-quality sugar beet varieties. Based on these tests, specific sugar beet varieties can be recommended to farmers and sugar enterprises, which can improve the local production and quality of sugar beet.

High yield and stability are crucial indicators of crop quality and must be considered during the breeding and evaluation of crop varieties (*Cao et al., 2023*). During sugar production, sugar content is as crucial as root tuber yield, as it influences the overall sugar yield (*Duan et al., 2014*). Regional variety experiments are typically used to evaluate GEI and variety adaptation patterns (*Studnicki et al., 2019*). These tests involve a comprehensive evaluation of the yield, adaptability, stress resistance, and quality of newly cultivated varieties in a specific region. GEI studies integrate stress resistance and quality parameters, allowing the selection of raw material varieties that meet the needs of the sugar industry (*Zhang et al., 2013*), and can thus guide the development of suitable varieties. Notably, accurate, reliable, and representative regional test data are crucial for performing comprehensive scientific analyses and evaluating plant varieties using regional tests.

Powerful statistical methods are essential for objectively and accurately evaluating GEI effects in crops. Prior to the development of the additive main effect and multiplicative interaction (AMMI) model, the most commonly used approaches were the two-factor ANOVA model, the Linear regression model, and the generalized linear model (GLM). These models can analyze both main effects and interaction effects using regional variety test data. However, their application in GEI evaluation is limited (*Guo et al., 2017*). Hence, the AMMI model is widely used for the analysis of regional experimental data. The AMMI model combines principal component analysis (PCA) with variance analysis to elucidate the interaction effect between a genotype and the environment. By separating the sum of several product terms from the interaction terms of the additive model, it can improve the accuracy of estimation. The AMMI model can be used to examine the major effects of genotypes and the environment, as well as GEI (*Chen et al., 2003*). Another tool called
the genotype main effect and genotype × environment interaction (GGE) biplot or the G+G×E model, which was developed by *Yan (2001)* has proven effective for evaluating performance parameters (*Bocianowski et al., 2019*). GGE biplots are generated from the matrix of original data and contain only two parts: the genotype main effect and the GEI effect. In this model, eigenvalue decomposition is performed simultaneously, and the principal component that explains the greatest amount of variation is represented and highlighted. GGE biplots can be used to examine the regional yield of test varieties, their regional adaptability, and the discriminating ability and representativeness of regional test sites. Hence, they can be employed to intuitively compare the performance of different varieties across various locations (*Akinwale et al., 2014*).

So far, both AMMI and GGE biplot models have been extensively applied for the analysis of high and stable yield in maize (*Bocci et al., 2020*), rice (*Lu et al., 2022*; *Wei et al., 2023*), oats (*Wei et al., 2021*), wheat (*Gao et al., 2008*; *Li et al., 2021*), sugarcane (*Wang et al., 2022*), and other crop varieties. In 2012, *Nei et al. (2012)* applied the AMMI model to conduct a stability analysis of root tuber yield, sugar content, and sugar yield in 16 sugar beet varieties. Further, they comprehensively evaluated the tested varieties based on the phenotypic values of the two traits and the corresponding variety stability parameter (Dg). In 2020, *Shao et al. (2020)* used the AMMI and GGE models, as well as the AMMI stable value (ASV), to evaluate 49 beet genotypes across four different geographical locations. They identified genotypes with a stable root yield, sugar content, and white sugar yield while also elucidating the discriminative ability of each regional environment. However, AMMI- and GGE model-based studies involving regional experiments for the introduction of sugar beet varieties to north and northeast China have been rather limited to date.

In this study, we utilized the AMMI model and GGE biplots to jointly examine the yield, regional adaptability, representativeness, and discriminativeness of 16 sugar beet varieties across seven experimental sites in north and northeast China. We analyzed the yield and quality traits of the sugar beet varieties, providing a scientific basis for the selection of sugar beet varieties with a high and stable yield in these regions.

## MATERIALS AND METHODS

### Test varieties and sites

Tables 1 and 2 show the main features of the tested varieties and sites. The experiments were conducted in seven pilot areas across north and northeast China (Fig. 1). These included two areas in Hulan District in Harbin City, Heilongjiang Province (including one "disease" area and one "non-disease" area; in the "disease" area, the experimental site had undergone crop rotation for many years, and the soil had an even distribution of pathogenic bacteria, which are the main causes of root rot and brown spot in beet); Yi'an County in Qiqihar City; Fanjiatun Town in Gongzhuling City, Jilin Province; Hongxinglong Farm in Jiamusi City; Zhangbei County in Zhangjiakou City, Hebei Province; and Jalaid Banner in Xingan League, Inner Mongolia Autonomous Region.

**Table 1 Description of sugar beet varieties examined and corresponding pilot codes.**

| Variety code | Variety name | Testing site code | Testing site |
| --- | --- | --- | --- |
| G1 | KWS 7748 | E1 | Hulan District (disease area) |
| G2 | KWS 1197 | E2 | Yi'an County |
| G3 | KWS 4207 | E3 | Gongzhuling City |
| G4 | KWS 1130 | E4 | Hulan District (non-disease area) |
| G5 | KWS 1132 | E5 | You'yi County |
| G6 | KWS 4254 | E6 | Zhangbei County |
| G7 | KWS 4253 | E7 | Jalaid Banner |
| G8 | KWS 4290 | | |
| G9 | KWS 3601 | | |
| G10 | KWS 4018 | | |
| G11 | KWS 4022 | | |
| G12 | KWS 4012 | | |
| G13 | KWS 4009 | | |
| G14 | KWS 4027 | | |
| G15 | KWS 4167 | | |
| G16 | KWS 4165 | | |

**Table 2 Agroclimatic characteristics of the locations studied in this research.**

| Province | Location | Longitude | Latitude | Altitude (m) | Minimum temperature (°C) | Maximum temperature (°C) | Average annual rainfall (mm) |
| --- | --- | --- | --- | --- | --- | --- | --- |
| Heilongjiang | Hulan District (disease area) | 126°58′ | 45°90′ | 118 | −7 | 33 | 406.4 |
| Heilongjiang | Yi'an County | 125°10′ | 47°01′ | 210 | −10 | 38 | 455.1 |
| Jilin | Gongzhuling City | 125°05′ | 43°43′ | 204 | −6 | 36 | 1,304.1 |
| Heilongjiang | Hulan District (non-disease area) | 126°58′ | 45°90′ | 118 | −7 | 33 | 406.4 |
| Heilongjiang | You'yi County | 125°05′ | 43°43′ | 204 | −3 | 39 | 670.4 |
| Hebei | Zhangbei County | 114°46′ | 41°40′ | 1,342 | −8 | 35 | 211.8 |
| Neimenggu | Jalaid Banner | 123°03′ | 46°41′ | 173 | −6 | 36 | 461.8 |

## Experimental design

A completely randomized block design was adopted for each experiment. Each variety of sugar beet was planted in four identical sets of two rows (each measuring 10 m in length), with the ridge spacing set at 0.6 m. The total area of each plot was 633.6 m², and the planting density was approximately 82,500 plants/ha. All other cultivation management procedures were conducted in accordance with experimental requirements.

## AMMI and GGE biplot analysis

In the first phase, normal ANOVA was used to estimate the main effects of the genotype and environment. In the second phase, the interaction residuals (residuals retained after
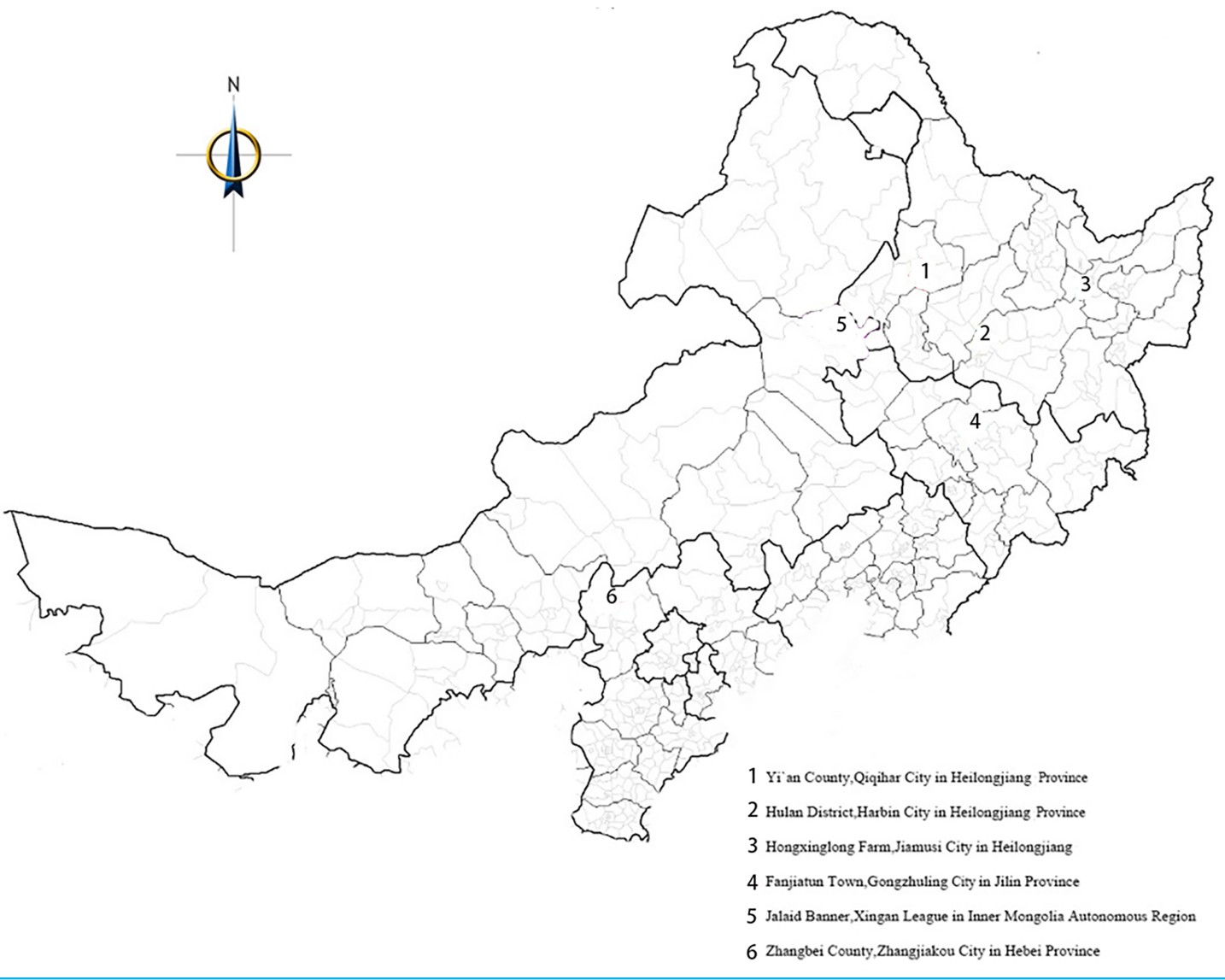

1 Yi'an County,Qiqihar City in Heilongjiang Province
2 Hulan District,Harbin City in Heilongjiang Province
3 Hongxinglong Farm,Jiamusi City in Heilongjiang
4 Fanjiatun Town,Gongzhuling City in Jilin Province
5 Jalaid Banner,Xingan League in Inner Mongolia Autonomous Region
6 Zhangbei County,Zhangjiakou City in Hebei Province

**Figure 1  Test site distribution map.**                               

removing the main effects) were subjected to PCA to obtain a clear understanding of the relationship between the genotype and the environment. This facilitated the evaluation of each variety's performance in a different environment, enabling the selection of the best-performing varieties.

GGE biplots are commonly employed for analyzing data from multi-environment trials (METs) of crops (*Hasani, Hamze & Mansori, 2021*). The GGE biplot method enables the visualization of both genotype main effects (G) and genotype-environment effects (GE) in a data table during the analysis of genotype-environment data. In this study, the GGE biplot mean matrix was generated based on the environment and was composed of singular value decomposition (SVD) as the primary component. Subsequently, a graph was produced by utilizing the initial two principal component scores (PC1 and PC2). GGE

biplot analysis was conducted using the GGEBiplot-GUI package (*Yan, 2001*) in R software Version 4.1.3.

## RESULTS

### Yield analysis of sugar beet varieties using the AMMI and GGE models

*AMMI analysis*

The combined ANOVA and AMMI analysis is shown in Table 3. The environmental main effect accounted for 74.25% of the total treatment variance. Conversely, the GEI contributed to only 13.99% of the overall variation. The proportion of sum-of-squares (SS) attributed to genotypes (varieties) was relatively low, accounting for merely 2.93% of the total value. However, significance tests revealed that all three components were highly significant ($P < 0.01$). This indicates that the differences in root yield observed in regional experiments were primarily due to variations in environmental conditions and the interaction between genotypes and the environment.

Further analysis based on the AMMI model allowed for a comprehensive examination of the interaction effect. Three IPCAs were extracted, and all of them were significant. IPCA1, IPCA2, and IPCA3 contributed to 49.88%, 18.26%, and 12.44% of the interaction SS, explaining 80.58% of the GEI. The substantial influence of these principal component axes indicates that the genotypes produced differential responses to different environmental conditions (*Pourdad & Moghaddam, 2013*). The analysis further confirmed that the AMMI model can effectively elucidate the interaction effects between genes and the environment. The AMMI IPCA1 and IPCA2 scores of sugar yield for each genotype and the corresponding ASVs are shown in Table 4. Smaller Dg values for a specific variety indicate greater stability. Based on the ASV ranking, KWS4207 (G3) displayed the highest stability, followed by KWS1130 (G4), KWS4009 (G13), KWS7748 (G1), and KWS4167 (G15). Meanwhile, KWS4018 (G10) exhibited the poorest stability. However, only KWS4009 (G13), KWS4167 (G15), and KWS1197 (G2) were identified as stable varieties with a high yield.

Table 5 demonstrates the yield discrimination ability of the different regions based on the AMMI model. De is the stability parameter, and higher De values indicate a stronger ability for discrimination. Fanjiatun Town (E3) and Jalaid Banner in Xingan League (E7) demonstrated the best discriminating ability. Conversely, the discriminating ability of Hulan District (disease area) (E1) and Yi'an County (E2) was weak.

*GGE biplot analysis*

The PCA of genotype (G) and GEI variation was conducted using the GGE biplot model. Here, PC1 and PC2 accounted for 42.12% and 21.43% of the total variation, respectively, explaining 63.55% of the effects of genotype and GEI. To assess the regional adaptability of the varieties, a polygon view map of the GGE biplot was prepared, and the best-performing genotypes in each environment were identified. This provided a concise summary of the GEI model (Fig. 2). The same variety showed diverse performance characteristics across different areas, reflecting its adaptability to specific regions. In the biplot, polygons were generated by connecting the markers representing the varieties farthest from the origin; all

**Table 3 ANOVA and AMMI analysis of yield in different sugar beet varieties.**

| Source of variance | Degrees of freedom (df) | Sum of squares (SS) | Mean square (MS) | F-value (F) | Proportion in the total SS |
|---|---|---|---|---|---|
| Total | 335 | 126,381,988,247.13 | – | – | – |
| Treatments | 14 | 126,381,988,247.13 | 71,433,764.32 | 1.48 | – |
| Genotypes | 15 | 3,703,204,839.47 | 246,880,322.63 | 5.10 | 2.93 |
| Environments | 6 | 93,832,630,263.30 | 15,638,771,710.55 | 323.29 | 74.25 |
| Genotypes × environments | 90 | 17,687,559,500.18 | 196,528,438.89 | 4.06 | 13.99 |
| PCA1 | 20 | 8,821,988,993 | 441,099,450 | 9.12 | 49.88 |
| PCA2 | 18 | 3,229,404,819 | 179,411,379 | 3.71 | 18.26 |
| PCA3 | 16 | 2,199,615,513 | 137,475,970 | 2.84 | 12.44 |
| Error | 210 | 10,158,520,943.71 | 48,373,909.26 | – | – |

**Table 4 Mean sugar yield (kg/ha), AMMI stability values (ASVs), and rankings of the 16 genotypes tested across seven environments.**

| Variable | Mean yield (kg/ha) | Mutual principal components | | | | Stability parameter (Dg) |
|---|---|---|---|---|---|---|
| | | Deviation | IPCA1 | IPCA2 | IPCA3 | |
| G1 | 48,311.33 | −4,220.63 | −16.31 | 36.42 | 0.00 | 9.69 |
| G2 | 60,216.07 | 7,684.12 | −9.12 | 66.99 | 0.00 | 11.34 |
| G3 | 53,685.01 | 1,153.05 | 9.66 | −32.79 | 0.00 | 7.67 |
| G4 | 53,324.49 | 792.53 | 27.63 | −38.05 | 0.00 | 7.87 |
| G5 | 48,296.61 | −4,235.34 | −101.9 | 47.67 | 0.00 | 21.94 |
| G6 | 55,817.99 | 3,286.04 | 33.87 | −64.63 | 0.00 | 12.25 |
| G7 | 46,913.43 | −5,618.52 | 67.22 | −47.22 | 0.00 | 18.33 |
| G8 | 51,085.21 | −1,446.75 | 32.78 | 50.67 | 0.00 | 12.51 |
| G9 | 56,159.73 | 3,627.78 | −57.80 | −11.05 | 0.00 | 15.25 |
| G10 | 51,011.20 | −1,520.76 | 108.56 | 13.00 | 0.00 | 22.51 |
| G11 | 48,894.44 | −3,637.52 | 57.75 | 71.81 | 0.00 | 20.57 |
| G12 | 52,214.27 | −317.69 | −62.68 | −76.86 | 0.00 | 16.49 |
| G13 | 54,120.12 | 1,588.17 | −9.08 | 31.96 | 0.00 | 8.19 |
| G14 | 52,645.47 | 113.51 | −13.97 | −20.24 | 0.00 | 13.57 |
| G15 | 53,933.26 | 1,401.30 | 40.91 | −19.19 | 0.00 | 11.03 |
| G16 | 53,882.66 | 1,350.71 | −107.5 | −8.49 | 0.00 | 20.25 |

other test varieties were encompassed within this polygon. The map also contained a set of perpendicular lines that divided the biplot into sectors, which represented regional clusters. The variety located at the apex of each sector's polygon was deemed the best variety in that particular region and was considered to exhibit the highest yield potential.

As shown in Fig. 2, the pilot regions were distributed across three distinct sectors in the GGE biplots. Yi'an County (E2) and Zhangbei County (E6) formed an ecological zone in which KWS4018 (G10) displayed the best performance. Meanwhile, KWS1130 (G2)

**Table 5 Evaluation of discriminating power for yield based on AMMI analysis.**

| Variable | Mean yield (kg/ha) | Mutual principal components | | | | Stability parameter (De) |
|---|---|---|---|---|---|---|
| | | Deviation | IPCA1 | IPCA2 | IPCA3 | |
| E1 | 46,440.3409 | −6,091.6145 | −203.310 | 17.0367 | 0 | 15.46 |
| E2 | 63,410.2564 | 10,878.301 | 17.8023 | −3.5387 | 0 | 13.43 |
| E3 | 40,332.6923 | −12,199.263 | 32.8329 | 125.5524 | 0 | 20.34 |
| E4 | 68,593.75 | 16,061.7946 | −1.9551 | −128.893 | 0 | 19.00 |
| E5 | 69,983.9744 | 17,452.019 | 48.9372 | −9.7593 | 0 | 17.49 |
| E6 | 58,812.5 | 6,280.5446 | 94.7618 | −4.3608 | 0 | 18.99 |
| E7 | 20,150.1736 | −32,381.782 | 10.9311 | 3.9629 | 0 | 22.58 |

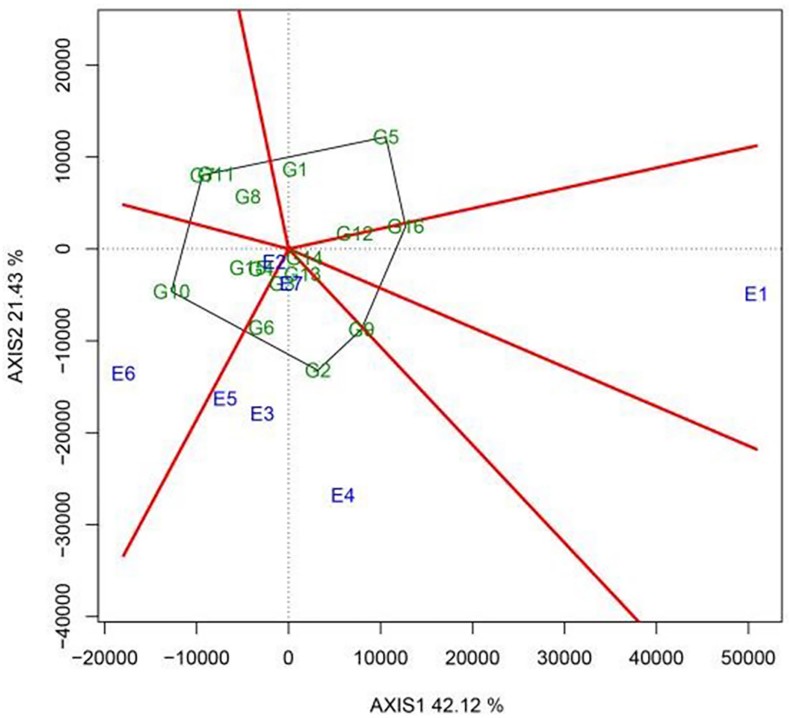

**Figure 2 Which-Won-Where polygon view of the GGE scatter biplot for yield in 2022 showing sugar beet genotypes with best performance in each environment.**

showcased excellent performance in Jalaid Banner in Xingan League (E7); Hongxinglong Farm (E5); Fanjiatun Town (E3); and Hulan District (non-disease area) (E4). Finally, in the ecological region consisting of Hulan District (disease area) (E1), KWS4165 (G16) exhibited the best performance.

Average environment coordinates in a GGE biplot are suitable indicators of stability. Moreover, the straight line passing through the origin of the axis, marked by an arrow,

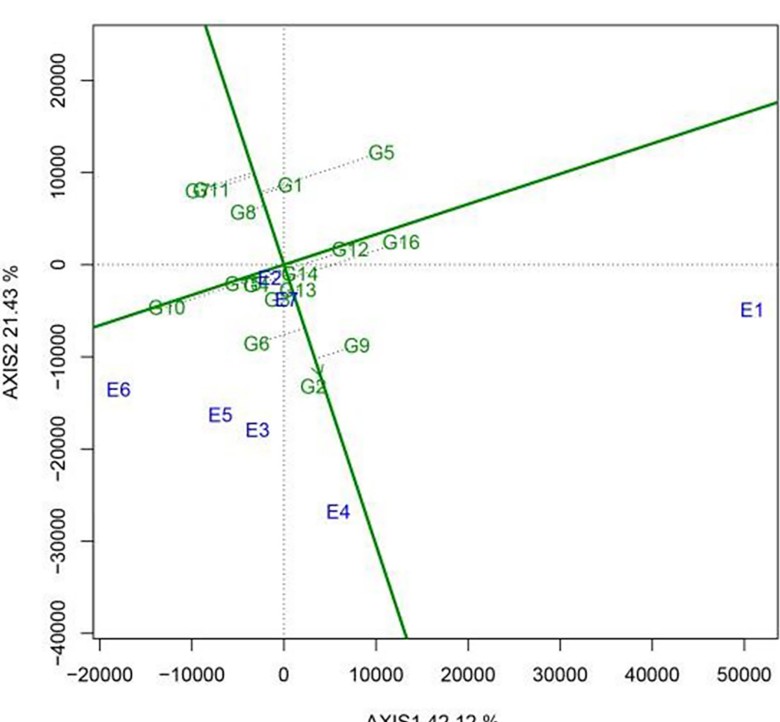

**Figure 3** The high yield and stable yield feature plot of the GGE scatter biplot for 2022 production shows the yield and stability in each variety.

represents the average environment axis, and the direction of the arrow indicates the positive direction of this axis. The solid horizontal axis, known as the average environment axis, represents the average environment, while the origin is associated with stability. The proximity of a variety to the average environment axis indicates its level of stability. Meanwhile, the solid vertical axis represents the mean value of yield for each genotype. Varieties located closer to the arrow on the solid vertical axis have higher yields. Consequently, genotypes with above-average yields are positioned toward the left side of the axis, while those with lower yields are located on the right.

A GGE biplot was employed to generate a functional chart that shows the varieties with a "high yield and stable yield." Figure 3 illustrates the results for the 16 sugar beet varieties. Among these varieties, KWS1197 (G2) achieved the highest level of stability, followed by KWS4009 (G13) and KWS4027 (G14). Meanwhile, KWS1132 (G5) exhibited the lowest level of stability. In terms of yield, KWS1197 (G2) exhibited the highest production, followed by KWS3601 (G9) and KWS4254 (G6). Based on comprehensive analysis, KWS1197 (G2) was identified as the best-performing variety, while KWS1132 (G5) showed the least satisfactory performance.

The discriminating power and representativeness of a pilot site are crucial factors that determine its effectiveness and suitability for trials. The "discriminating power and representativeness" function chart of the GGE biplot is useful for assessing these

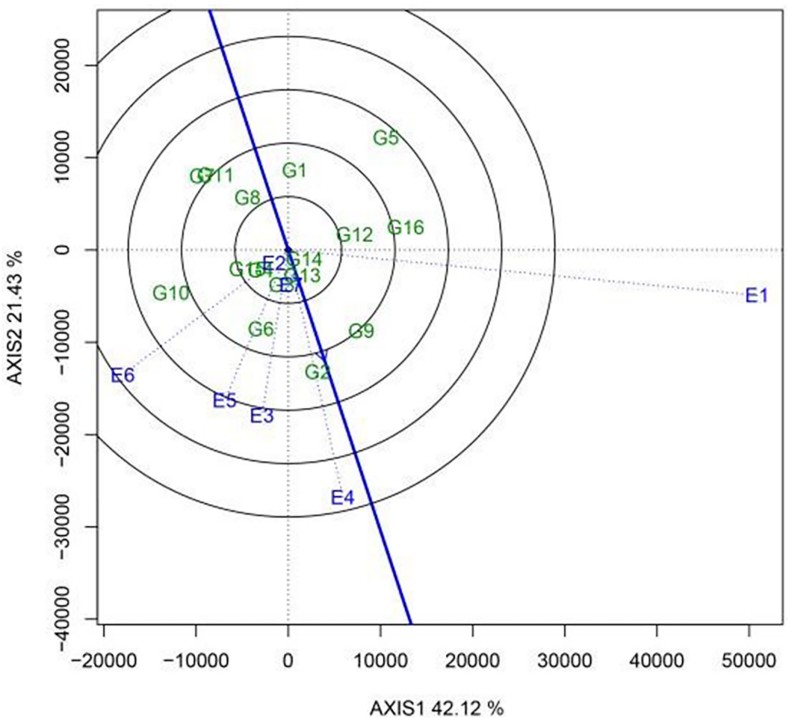

**Figure 4 The differentiating power and representative function maps of the GGE scatter biplot for 2022 production show the discriminating ability in each pilot.**

indicators. The arrow representing the "average environment axis" can be used to comprehensively evaluate the discriminating power and representativeness of a pilot site. The length of the environment vector, *i.e.*, the distance between the origin of the coordinates and the marker for the pilot site, represents the discriminating ability of the site. A longer vector signifies a stronger discriminating ability, whereas a shorter one is indicative of a weaker discriminating ability. Meanwhile, the representativeness of a regional pilot site is indicated by the angle between the environment vector of the pilot site and the forward mean environmental axis. A smaller angle indicates better representativeness, while a larger angle indicates poorer representativeness. An obtuse angle signifies that the regional environment is not suitable for conducting pilot tests.

The discriminating power and representativeness of the regions used for sugar beet pilots were examined using GGE biplots (Fig. 4). Hulan District (disease area) (E1); Hulan District (non-disease area) (E4); and Zhangbei County (E6) demonstrated a superior discriminating power. Meanwhile, Hulan District (non-disease area) (E4); Jalaid Banner in Xingan League (E7); Fanjiatun Town (E3); and Hulan District (disease area) (E1) exhibited a good discriminating ability but poor representativeness. Hulan District (non-disease area) (E4) displayed both good representativeness and discriminativeness, making it an ideal pilot site for the regional trial.

### Comparison of AMMI and GGE biplot results

The results in Tables 4 and 5 were compared to those in Table 6. The two analysis methods yielded slightly different results with regard to the yield of sugar beet varieties and the yield-discriminating power of the test sites. However, these differences were not significant.

According to the AMMI results, KWS1197 (G2), KWS4254 (G6), KWS3601 (G9), and KWS4009 (G13) ranked first, third, second, and fourth, respectively, in terms of yield. Meanwhile, GGE analysis placed the yield of these varieties in the first, third, second, and fifth positions, respectively. Given their high rankings, these four varieties were identified as high-yield varieties.

Meanwhile, KWS4009 (G13), KWS4207 (G3), KWS1130 (G4), and KWS1197 (G2) ranked third, first, second, and sixth in terms of stability, respectively, according to AMMI analysis. However, GGE stability rankings placed them at the second, fourth, fifth, and first positions. Nevertheless, these varieties showed consistently high average rankings and evidently exhibited exceptional stability. Owing to both their high yield and stability, KWS1197 (G2) and KWS4009 (G13) appeared to be suitable as priority planting options.

Hulan District (non-disease area) (E4) and Fanjiatun Town (E3) ranked third and second in terms of discriminating power, respectively, in the AMMI analysis. Meanwhile, GGE analysis ranked these sites as first and second, respectively, indicating their superior performance. Hence, these pilot sites were found to be the most discriminative. Overall, the results showed that both AMMI and GGE biplot analysis were reliable tools for the evaluation of high yield, stable yield, and pilot discriminating power.

## Sugar content analysis of sugar beet varieties using the AMMI and GGE models

### AMMI analysis

The sugar content of sugar beet was analyzed using joint ANOVA and AMMI analysis (Table 7). The SS of the main environmental effects accounted for 67.90% of the total treatment SS, whereas the SS of GEI accounted for 14.30% of the total value. The proportion of the genotype SS was only 5.30% ($P < 0.01$). The AMMI model was used to further explain the decomposition of the interaction SS, and two extremely significant principal component axes were obtained. The SS of IPCA1, IPCA2, and IPCA3 accounted for 44.12%, 20.81%, and 15.45% of the interaction SS, respectively. Together, they explained 80.38% of the GEI SS. The GEI were significant, indicating that different genotypes have different responses to different environments. Moreover, the analysis showed that the AMMI model can effectively analyze and explain the interaction effect between genes and the environment. AMMI analysis of the sugar content of the tested varieties (Table 8) showed that KWS4012 (G12) had the best stability, followed by KWS4165 (G16), KWS4018 (G10), and KWS4022 (G11). Further, KWS4027 (G14) and KWS4207 (G3) were the varieties with a high and stable sugar content.

The stability parameter De for the pilot sites was obtained using the AMMI model (Table 9). Hulan District (disease area) (E1) and Jalaid Banner in Xingan League (E7) were found to have the strongest discriminative ability for sugar content.

**Table 6 Comparison of yield based on AMMI analysis and the GGE biplot.**

| Variety | Fertility | | Stability | | Testing site | Discrimination | |
|---|---|---|---|---|---|---|---|
| | AMMI rank | GGE rank | AMMI rank | GGE rank | | AMMI rank | GGE rank |
| G1 | 14 | 13 | 4 | 8 | E1 | 6 | 3 |
| G2 | 1 | 1 | 6 | 1 | E2 | 7 | 7 |
| G3 | 7 | 4 | 1 | 4 | E3 | 2 | 2 |
| G4 | 8 | 8 | 2 | 5 | E4 | 3 | 1 |
| G5 | 15 | 14 | 15 | 14 | E5 | 5 | 5 |
| G6 | 3 | 3 | 7 | 11 | E6 | 4 | 6 |
| G7 | 16 | 16 | 12 | 12 | E7 | 1 | 4 |
| G8 | 11 | 12 | 8 | 6 | | | |
| G9 | 2 | 2 | 10 | 9 | | | |
| G10 | 12 | 11 | 16 | 15 | | | |
| G11 | 13 | 15 | 14 | 10 | | | |
| G12 | 10 | 9 | 11 | 13 | | | |
| G13 | 4 | 5 | 3 | 2 | | | |
| G14 | 9 | 6 | 9 | 3 | | | |
| G15 | 5 | 10 | 5 | 7 | | | |
| G16 | 6 | 7 | 13 | 16 | | | |

**Table 7 ANOVA and AMMI analysis of sugar content in sugar beet varieties.**

| Source of variance | Degrees of freedom (df) | Sum of squares (SS) | Mean square (MS) | F-value (F) | Proportion in the total SS |
|---|---|---|---|---|---|
| Total | 335 | 866.3652 | – | – | – |
| Treatments | 14 | 29.666 | 2.119 | 5.6524 | – |
| Genotypes | 15 | 45.8568 | 3.0571 | 8.1548 | 5.30 |
| Environments | 6 | 588.246 | 98.041 | 261.5238 | 67.90 |
| Genotypes × environments | 90 | 123.8708 | 1.3763 | 3.6714 | 14.30 |
| PCA1 | 20 | 54.647719 | 2.732386 | 7.29 | 44.12 |
| PCA2 | 18 | 25.778198 | 1.432122 | 3.82 | 20.81 |
| PCA3 | 16 | 19.142159 | 1.196385 | 3.19 | 15.45 |
| Error | 210 | 78.7256 | 0.3749 | – | – |

### GGE biplot analysis

As shown in Fig. 5, PCA of genotype (G) and GEI variation was conducted using the GGE biplot model. PC1 and PC2 accounted for 32.43% and 29.94% of the total variation, respectively, and could thus explain 62.37% of genotype and GEI effects. The pilot could be divided into two large ecological areas according to the sugar content of the tested varieties. The first ecological area was composed of Zhangbei County (E6); Hongxinglong Farm (E5); Jalaid Banner in Xingan League (E7); and Fanjiatun Town (E3). KWS4027 (G14)

**Table 8 Mean sugar content (%), AMMI stability values (ASV), and rankings of the 16 genotypes tested across seven environments.**

| Variable | Mean yield (kg/ha) | Mutual principal components | | | | Stability parameter (Dg) |
|---|---|---|---|---|---|---|
| | | Deviation | IPCA1 | IPCA2 | IPCA3 | |
| G1 | 15.53 | −0.1721 | −0.1813 | 0.721 | 0.1709 | 3.71 |
| G2 | 15.85 | 0.1455 | 0.6642 | −0.3432 | −0.5855 | 4.89 |
| G3 | 16.23 | 0.5293 | 0.4969 | −0.2176 | 0.036 | 3.36 |
| G4 | 16.09 | 0.3846 | 1.0114 | 0.1706 | 0.3409 | 5.92 |
| G5 | 15.82 | 0.115 | −0.8147 | −0.4298 | −0.4943 | 5.40 |
| G6 | 15.72 | 0.0169 | −0.3746 | −0.4743 | 0.6166 | 4.27 |
| G7 | 16.23 | 0.5284 | −0.5811 | 1.0048 | 0.2233 | 5.47 |
| G8 | 15.36 | −0.3476 | −0.0369 | −0.5924 | 0.2671 | 3.28 |
| G9 | 15.41 | −0.2974 | 0.1277 | 0.1623 | 0.4159 | 4.42 |
| G10 | 15.33 | −0.374 | −0.1179 | −0.2982 | −0.1068 | 2.55 |
| G11 | 15.43 | −0.2693 | 0.4441 | 0.0926 | 0.3256 | 3.13 |
| G12 | 15.34 | −0.3631 | −0.1014 | 0.1742 | −0.1763 | 2.06 |
| G13 | 15.38 | −0.3243 | 0.7387 | 0.2643 | −0.6063 | 5.04 |
| G14 | 16.52 | 0.8217 | −0.3879 | −0.4367 | 0.4489 | 3.66 |
| G15 | 15.52 | −0.1852 | −0.608 | 0.1795 | −0.53 | 4.40 |
| G16 | 15.49 | −0.2085 | −0.2794 | 0.023 | −0.346 | 2.45 |

**Table 9 Evaluation of discriminating power for sugar content based on AMMI analysis.**

| Variable | Mean sugar content (%) | Mutual principal components | | | | Stability parameter (De) |
|---|---|---|---|---|---|---|
| | | Deviation | IPCA1 | IPCA2 | IPCA3 | |
| E1 | 13.38 | −2.3233 | 0.9242 | −0.014 | 0.6612 | 9.05 |
| E2 | 16.28 | 0.5799 | 0.2741 | −0.659 | 0.2459 | 5.33 |
| E3 | 17.16 | 1.4574 | −0.3751 | −0.327 | 0.4399 | 4.44 |
| E4 | 14.39 | −1.3181 | 1.295 | 0.3306 | −0.6801 | 5.44 |
| E5 | 17.33 | 1.6276 | −0.7835 | −0.960 | −0.507 | 6.19 |
| E6 | 15.82 | 0.1169 | −0.5753 | 0.6916 | −0.8225 | 4.86 |
| E7 | 15.56 | −0.1404 | −0.7594 | 0.9382 | 0.6627 | 6.65 |

performed the best in this ecological area. Meanwhile, Yi'an County (E2); Hulan District (disease area) (E1); and Hulan District (non-disease area) (E4) constituted the second ecological area; the best-performing variety in this area was KWS1130 (G4).

As shown in Fig. 6, KWS4290 (G8) had the best stability, but its yield was poor. The yield was the highest in KWS4018 (G10), followed by KWS3601 (G9), KWS4012 (G12), KWS4027 (G14), and KWS4207 (G14). KWS4207 (G14) had good stability and high yield, and it was thus considered a suitable variety.

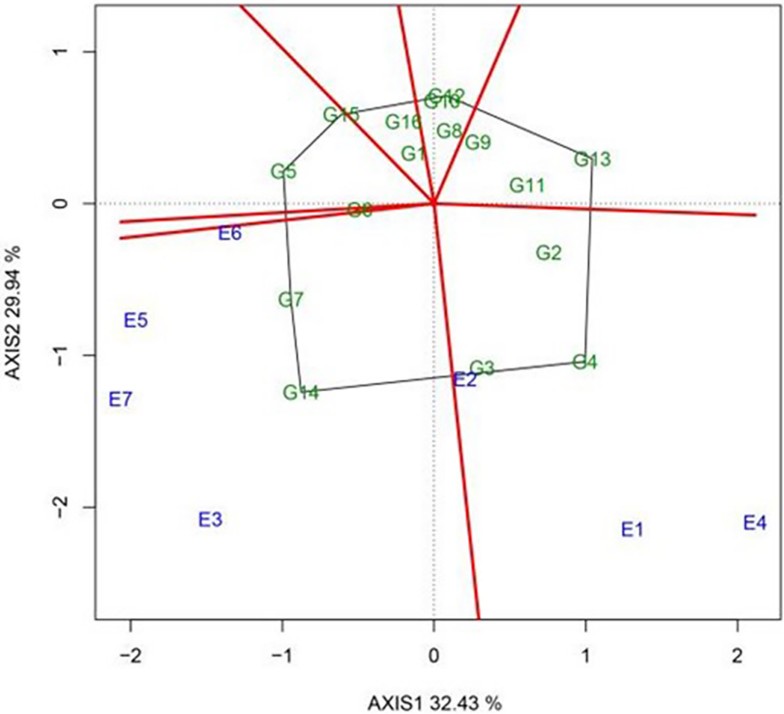

**Figure 5 Which-Won-Where polygon view of the GGE scatter biplot for sugar content in 2022 showing sugar beet genotypes with best performance in each environment.**

Figure 7 illustrates that Hulan District (non-disease area) (E4); Fanjiatun Town (E3); and Jalaid Banner (E7) of Xingan League had good discriminating ability for sugar content. Meanwhile, Fanjiatun Town (E3) and Yi'an County (E2) showed superior representativeness. Hence, Fanjiatun Town (E3) was the best pilot site.

### Comparison of AMMI and GGE biplot results

As shown in Table 10, KWS4207 (G3), KWS1130 (G4), and KWS4027 (G14) ranked second, fourth, and first, respectively, in terms of sugar content according to both AMMI and GGE analyses. Hence, these varieties were confirmed to have the highest sugar content. In terms of stability, KWS4012 (G12), KWS4290 (G8), and KWS4018 (G10) ranked first, fifth, and third according to AMMI, and second, first, and third according to GGE, respectively. Although these three varieties showed the best stability, their low sugar content made them poor candidates for planting. In contrast, KWS4027 (G14) and KWS7748 (G1) had a high and stable sugar content.

In terms of pilot discrimination, Jalaid Banner (E7) in Xingan League and Hulan District (disease area) (E1) ranked second and first according to AMMI and fourth and third according to GGE, respectively. Their discriminating power was higher than that of the other pilot sites on average, making them the most discriminative pilot sites.

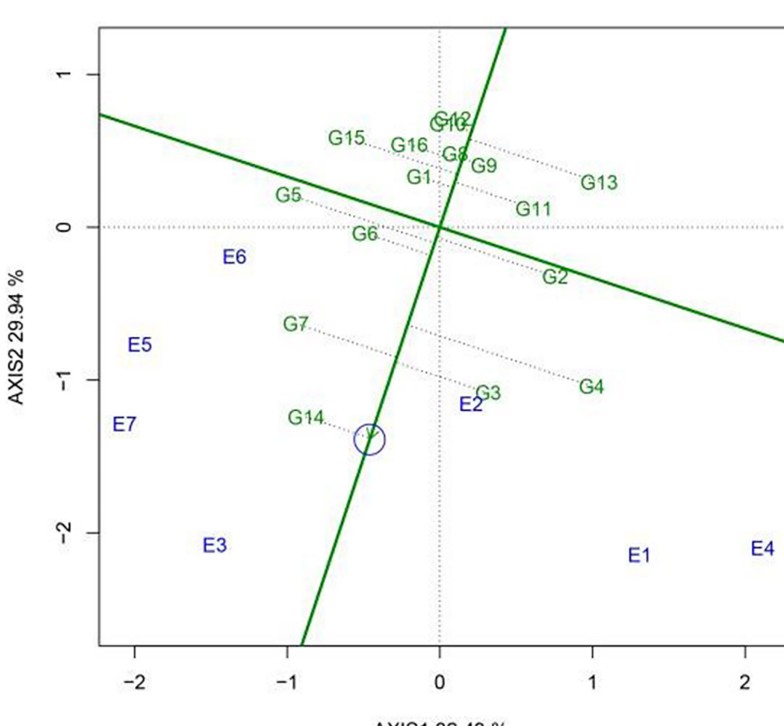

**Figure 6  The high yield and stable yield feature plot of the GGE scatter biplot for 2022 sugar content shows the yield and stability in each variety.**

## DISCUSSION

The yield and quality of sugar beet (sugar content) are usually affected by GEI. Specifically, the effects of GEI impede the breeding process by weakening the association between phenotype and genotype. This occurs due to the reduction in heritability and limits the selection of "good" genotypes across heterogeneous environments. Therefore, it is important to consider GEI while developing strategies to improve the yield of sugar beet (*Eltaher et al., 2021*). In addition to the inherent characteristics of the variety itself, key environmental factors such as soil and climate conditions can also influence the yield and quality of sugar beet (*Shanmuganathan et al., 2023*; *Bartsch et al., 2003*). Therefore, sugar enterprises and plant breeders prefer to cultivate sugar beet varieties with high quality and stability under different environmental conditions. Therefore, the present study investigated the yield of 16 sugar beet varieties across seven pilot sites in 2022. Comprehensive analysis showed that the environment had a higher influence on the overall variation in sugar beet yield than GEI, while genotype had the least influence. Moreover, environmental effects explained a large proportion of the yield differences. Therefore, to improve the yield of sugar beet, varieties that are suitable for growth in a specific environment must be selected. The combination of regional control and variety matching could be effective for improving the sugar content of sugar beet. This is consistent with the results obtained by *Khodadad & Mohammadreza (2018)* in a previous

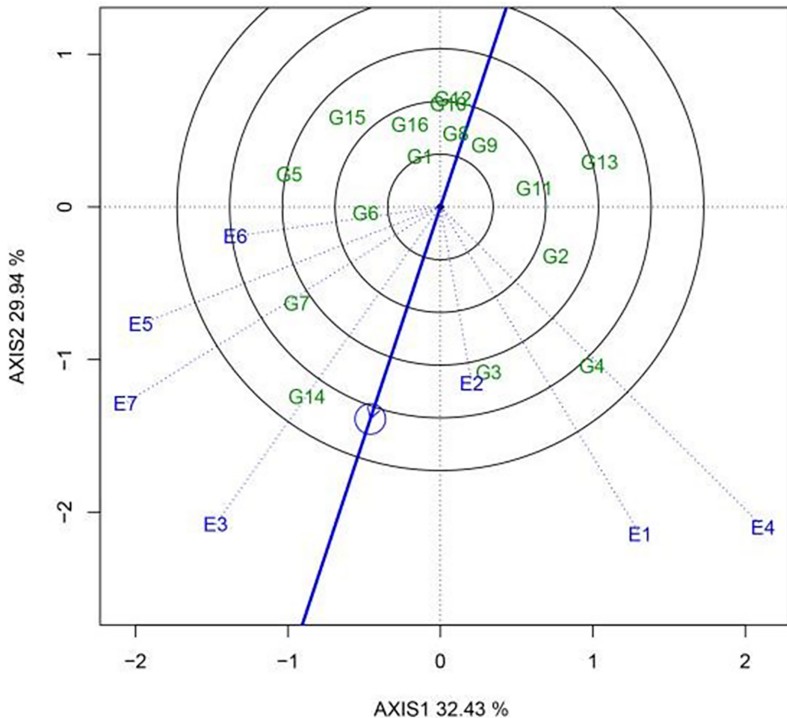

*Discriminating ability vs. representativeness*

**Figure 7  The differentiating power and representative function maps of the GGE scatter biplot for 2022 sugar content show the discriminating ability in each pilot.**

study. In the present study, there were several differences in the actual cultivation process across the seven test sites, largely owing to climate variations.

The AMMI model combines ANOVA and PCA to visualize the stability and adaptability of different varieties within each trial site (*Gao et al., 2008*). Moreover, it also helps in classifying total variation into genotype main effects, environmental main effects, and GEI, all of which are sources of variation and present different challenges and opportunities for agricultural researchers (*Gauch, 2006*). According to the AMMI model, most of the differences in sugar beet yield in this study could be explained by environmental variations, indicating that the environment had a very significant influence on yield traits. Notably, if the yield ranking of a genotype does not change across multiple environments, that is, GEI are absent or minimal, the variety is considered to demonstrate universal adaptation (*Baker, 1998*).

High and stable yield is crucial for determining whether a variety is appropriate for widespread production across a large area (*Ye et al., 2020*). The superior yield exhibited by high-yield varieties, coupled with the capacity of stable-yield varieties to adapt to environmental changes through self-regulated gene expression, leads to relatively stable growth and development (*Frutos, Galindo & Leiva, 2014*). In this context, the use of the AMMI model and GGE biplots provides a theoretical basis for the rational development of new varieties and their accelerated adoption.

**Table 10 Comparison of sugar content based on AMMI analysis and the GGE biplot.**

| Variety | Fertility | | Stability | | Testing site | Discrimination | |
|---|---|---|---|---|---|---|---|
| | AMMI rank | GGE rank | AMMI rank | GGE rank | | AMMI rank | GGE rank |
| G1 | 8 | 8 | 8 | 5 | E1 | 1 | 3 |
| G2 | 5 | 7 | 12 | 13 | E2 | 5 | 2 |
| G3 | 2 | 2 | 6 | 10 | E3 | 7 | 1 |
| G4 | 4 | 4 | 16 | 16 | E4 | 4 | 5 |
| G5 | 6 | 6 | 14 | 15 | E5 | 3 | 6 |
| G6 | 7 | 5 | 9 | 7 | E6 | 6 | 7 |
| G7 | 3 | 3 | 14 | 11 | E7 | 2 | 4 |
| G8 | 14 | 13 | 5 | 1 | | | |
| G9 | 12 | 12 | 11 | 4 | | | |
| G10 | 16 | 15 | 3 | 3 | | | |
| G11 | 11 | 9 | 4 | 9 | | | |
| G12 | 15 | 16 | 1 | 2 | | | |
| G13 | 13 | 14 | 13 | 14 | | | |
| G14 | 1 | 1 | 7 | 6 | | | |
| G15 | 9 | 10 | 10 | 12 | | | |
| G16 | 10 | 11 | 2 | 8 | | | |

The differences between the results of the AMMI model and GGE biplots can partly be explained by the differences in the model assumptions and accuracies of these approaches. GGE biplots and AMMI analysis are based on different assumptions and prediction models; hence, their accuracy, reliability, and results can be different. For example, the GGE biplot assumes that both environment and genotype contribute to data variation, while the AMMI model assumes that environment contributes to the main variation components and genotype contributes to only some variation components. In our study, the GGE biplot and AMMI plots explained a high percentage of variance, which indicates the reliability of our results. From the perspective of cultivar stability and adaptability, the results obtained through these two approaches showed good consistency. Hence, these two methods could illustrate how different varieties behave in different environments and provide valuable information for the selection of the right variety for a specific region. These two statistical analysis methods can complement each other, overcoming the shortcomings of a single analysis model. Hence, their combination can further enhance the accuracy and reliability of the final conclusions and provide the necessary theoretical basis for screening new crop varieties with a high and stable yield. In this study, the rankings obtained based on the GGE biplot and AMMI models were calculated, and their average values were used as the final results.

Overall, the analysis of the interaction between sugar beet genotype and environment has broad application prospects. It can not only guide the breeding and cultivation of sugar beet but also contribute to the assessment of environmental adaptability and the development of sustainable agriculture. The popularization and application of this

analytical approach could improve the yield and quality of sugar beet and promote the sustainable development of agriculture. However, the present study also had certain limitations, which should be addressed in the future by obtaining data across multiple years and selecting more varieties for assessment to improve the stability and reliability of the results.

## CONCLUSIONS

This study demonstrates the importance of GEI and stability analysis for evaluating yield and quality traits (sugar content) in sugar beet across different environmental conditions. The high contribution of environment-related SS, as observed through both the GGE biplot and AMMI methods, indicates that environmental factors contribute significantly to the variation in sugar yield and sugar content. Our stability and adaptability analysis based on the AMMI and GGE biplot models showed that KWS4207 (G3) has a high and stable yield and quality, respectively. It also indicated that the most discriminative and representative pilots were those in Jalaid Banner (E7) in Xingan League, Inner Mongolia Autonomous Region. In future studies, more rigorous and comprehensive experimental designs, data analysis, and experimental quality control will be warranted. Further, corresponding improvements and adjustments according to on-ground conditions will also be required.

### Funding

This work was supported by the National Sugar Industry Technical System Decomposition Project "Research on Sugar Beet Ferti-lization Decision-making Model Based on UAV Remote Sensing Technology" No. CARS-170202; Research on soil improvement and carbon emission in black soil area by returning filtered mud from agricultural waste sugar plant to field, Heilongjiang Department of Ecology and Environment No. HST2022TR003. The funders had no role in study design, data collection and analysis, decision to publish, or preparation of the manuscript.

### Grant Disclosures

The following grant information was disclosed by the authors:
National Sugar Industry Technical System Decomposition Project "Research on Sugar Beet Ferti-lization Decision-making Model Based on UAV Remote Sensing Technology": CARS-170202.
Heilongjiang Department of Ecology and Environment: HST2022TR003.

### Competing Interests

The authors declare that they have no competing interests.

### Author Contributions

- Xinwang Dang analyzed the data, prepared figures and/or tables, and approved the final draft.

- Xiaohang Hu conceived and designed the experiments, performed the experiments, analyzed the data, prepared figures and/or tables, and approved the final draft.
- Yahuai Ma conceived and designed the experiments, performed the experiments, analyzed the data, authored or reviewed drafts of the article, and approved the final draft.
- Yanli Li performed the experiments, authored or reviewed drafts of the article, and approved the final draft.
- Wenliang Kan conceived and designed the experiments, performed the experiments, authored or reviewed drafts of the article, and approved the final draft.
- Xinjiu Dong conceived and designed the experiments, authored or reviewed drafts of the article, and approved the final draft.

## Data Availability

The raw measurements are available in the Supplemental Files.

## Supplemental Information

Supplemental information for this article can be found online at http://dx.doi.org/10.7717/peerj.16882#supplemental-information.

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
