# Peer review of "AMMI and GGE biplot analysis for genotype × environment interactions affecting the yield and quality characteristics of sugar beet"

_PeerJ, doi:10.7717/peerj.16882_

## Round 0.1 · original submission · Major Revisions

There is a need to fully revise the article, especially considering the queries of reviewer 3.

Reviewer 1 ·

Basic reporting

The manuscript demonstrates exemplary technical writing with a notable absence of discernible technical errors. However, there is a need for a thorough review and alignment of references to adhere meticulously to the journal's stipulated guidelines concerning both in-text citations and the reference list. Additionally, we recommend a comprehensive reassessment of the data presentation and analysis, as outlined in the comments provided within the manuscript.

Experimental design

well explained; further comments are mentioned in manuscript

Validity of the findings

Justification for conducting the analysis across two distinct years necessitates elucidation. Furthermore, a rationale for favoring individual annual analyses over a combined pooled analysis is warranted. Moreover, an articulation of the specific supplementary insights garnered from the individual annual analyses vis-à-vis the pooled analysis is essential.

Additional comments

Refer to the manuscript for detailed comments.

Annotated reviews are not available for download in order to protect the identity of reviewers who chose to remain anonymous.

·

Basic reporting

• English language used in the manuscript is clear but some repetitions of sentences and few grammatical mistakes are found so it is suggested to get read by some language experts for improvement.
• Many issues have found regarding references used by the authors in its font and format. It should be checked with proper care in the whole manuscript. Make it proper according to journal guidelines.
Eg: In line no. 89,105, “et al” used in the script should be in italics.
Line no. 97, 101“.,” should be used after et al.
Line no. 64,97,432,434, basic background formatting should be done properly.
Line no. 75, Zhang et al., 2013 should be used instead of all names.
Line no. 82, 102, full stop between “et al” is never used.
Line no. 99-100, either put Yan et al., 2000; Zobel 100 et al., 1988 in different brackets or all in one bracket for maize.
Line no. 189, references should in proper font as used by journal guidelines.
Line no. 396-397: continuity in sentence is missing.
Line no. 497: full stop should be used after end of sentence.

Experimental design

• Original research.
• Figure 1 used in the manuscript is not clear. Please use a clear image.
• Section 2.1 only has description of sites but not varieties so title should be revised.
• Line no. 130-131 is repeated earlier also, so please avoid repetition of sentences and use good professional English language.
• Line no. 134, meter square should be written as m2.
• In line no. 105, stability parameter is denoted by ‘Di’ and in line no. 202 onwards, it is denoted by ‘De’. Please use appropriate abbreviation.
• Line no. 202-205: In year 2021, only the stronger discriminating environments are written. Also, write the weaker ones.
• Line no. 202-213; 360-363: Mention the table no. from where the data in results are explained or the table is missing.
• Line no. 235-244: Mention the figure number for which explanation is given. Once check details in figure too for “arrow” and “small circle”.

Validity of the findings

• All the data provided are robust and statistically sound.
• Statistical analysis done for sugar content should be checked once more properly.
• Line no. 331: In table 7, sum of squares in total proportion is missing.
• Results explained from line no. 331-335 do not match with the table 7. Check the ANOVA table for sugar content analysis.
• Results explained from line no. 346-349 do not match with the table 8 (2021).
• Check sequence of environments mentioned for discriminating ability in line no. 355(weak) and 357(strong).
• Discussion part should be explained more properly with recent research done. Mostly used references in the manuscript are old and should be updated with relevant 2-3 years of research.
• Give a concert conclusion along with prospects and limitations of study.

Additional comments

Please go through the whole references used in the manuscript. Proper formatting and font style should meet the standards of journal. More number of genotypes are needed for good validation of results.

Reviewer 3 ·

Basic reporting

Article is written in well manner, some repetition has observed between introduction and result-discussion section which need to be rectify

Title may be modify as suggested

Experimental design

Experimental design is ok but the genotypes in both years are different hence two years results cant be compared better you make single article on either 2021 or 2022.

Analysis part is acceptable

Validity of the findings

See attached file for more details

Additional comments

Line 2 to 4 title should be
AMMI and GGE Biplot Analysis for Genotype × Environment Interactions Affecting the Yield
and Quality Characteristics of Sugar Beet

line 30 use comma instead of and word
line 38 mention total how much variation describe by two PC i.e 95.23% in 2021 and 80.58% in 2022
line 43: add word environment.
Lone 51: reduce introduction portion by avoiding repetition
Line 52: mention scientific name and its genetics (chromosome number and family etc.)
Line 98 to 102 reference of crops are irrelevant because in line 103 and 104 already reference available in relevant crops so remove it.
Line 118: You have mention 16 varieties but as per table 1 there is name of varieties different in both year 2021 and 2022. Weather they are different for both year or only coding is different so clarify and if they are different then we can’t pool the data for two years.

Line 352 to 358: here, in table 9, You have mention about year 2021 and 2022 but in R&D, description, genotype discriminating environment are different for both year which may be due to different genotypes you have choose for both years.
Line 434 and 435: total 32 varieties/genotypes tested in 2021 and 2022.
Line 452 to 454; in introduction you have clearly mention about advantage of AMMI and biplot so may not compare this with other methods in discussion.
Line 465: where is real two-year data? Here genotype is different in both years.
Line 477: you have mention different varieties in both year? How can be compare it?
Line 489 to 491 & 501 to 504 sentences are similar so slightly modified it and present here.
Overall, data presentation is good but there are so many places repetition in sentences are observed so omit it.
Mention only precise results only some place you have given clarity/reason about it which make paper lengthy and bulky so avoided it.

---

## Round 0.2 · Minor Revisions

Revise article considering the comments.

Reviewer 1 ·

Basic reporting

Line 121: Which disease are you concerned about? because there is no land where there is zero disease inoculum. So as you are differentiating between diseased and non-diseased, give clear information about a particular disease (its name). soil born or air born ?
Lines 139 and 140 are not necessary to write, as you almost delete all the details and information about statistical analysis from material and method.
line 310: (Bartsch D et al., 2003). Follow the guidelines for writing references.
The discussion part should be more elaborate, citing' other works and their outcomes.
Table 4 is not appropriate. There are no details regarding ASV values or ranking in the table. Change it.

Experimental design

Appropriate

Validity of the findings

Ok

·

Basic reporting

English used in the literature meets the standard and the necessary corrections have been done properly.

Experimental design

The recommended corrections in the tables and figures have been incorporated.

Validity of the findings

Suggested corrections have been done and discussion part is elaborated as per recommendation.

---

## Round 0.3 · Minor Revisions

The revised article has been approved by the reviewer and hence may accepted for publication from a scientific standpoint.

However the writing of the current version is not acceptable for publication since the grammar is erroneous, and some parts cannot be understood. Examples: First sentence of the abstract "Sugar beet is an important sugar crop globally." What is a layout test for new varieties? etc. Revise the article taking the help for language service. During resubmission provide evidence for a professional proofreading certificate.

**Language Note:** The Academic Editor has identified that the English language must be improved. PeerJ can provide language editing services - please contact us at [email protected] for pricing (be sure to provide your manuscript number and title). Alternatively, you should make your own arrangements to improve the language quality and provide details in your response letter. – PeerJ Staff

Reviewer 1 ·

Basic reporting

All the suggestions were incorporated, and the manuscript is ready to be accepted.

Experimental design

.

Validity of the findings

.

Additional comments

.

---

## Round 0.4 · accepted · Accept

The language of the article is improved and a certificate of professional proofreading has also been attached; the article has been accepted for publication.